# Spatial development of planar and axisymmetric wakes of porous objects under a pressure gradient: a wind tunnel study

Wessel van der Deijl[1], Martín Obligado[1,2], Stéphane Barre[1], and Christophe Sicot[3]

[1]Univ. Grenoble Alpes, CNRS, Grenoble INP, LEGI, Grenoble, 38000, France
[2]Univ. Lille, CNRS, ONERA, Arts et Métiers ParisTech, Centrale Lille, UMR 9014 –LMFL – Laboratoire de Mécanique des Fluides de Lille – Kampé de Feriet, Lille, F-59000, France
[3]Institut Pprime - UPR 3346, CNRS-ENSMA-Université de Poitiers, Futuroscope-Chasseneuil, 86360, France

**Correspondence:** Martín Obligado (martin.obligado@centralelille.fr)

**Abstract.**

We report an experimental study on the effect of a constant adverse pressure gradient on the spatial evolution of turbulent wakes generated by different objects. A porous disk, designed to mimic the wake of a horizontal axis wind turbine, and a porous cylinder, whose wake matches that of a vertical axis wind turbine, were tested in a wind tunnel for Reynolds numbers (based on the generator diameter) in the range of $2.6 \times 10^5$ to $3.9 \times 10^5$. Experiments were conducted between 1 and 7 diameters downstream of the disk and from 2 to 12 diameters downstream of the cylinder.

We find that the effect of adverse pressure gradient is significant in all cases, resulting in larger velocity deficits and wider wakes. Moreover, these variations are stronger for the cylinder-generated wake. We also find that current analytical models for wakes evolving in pressure gradients, developed from momentum conservation, satisfactorily fit our data. Our results provide a benchmark case that will contribute to improving energy harvesting in cases where pressure gradients are relevant, such as in wind plants installed over complex topographies and tidal stream generators.

## 1 Introduction

Turbulent flows play a relevant role in several environmental and industrial situations. For instance, within wind farms, the energy loss in the wake of a wind turbine reduces the available power for downstream turbines. Therefore, in order to minimize these losses, and design an optimal wind farm layout, it essential to accurately model the turbulent wakes of wind turbines (Neunaber et al., 2022c; Kadum et al., 2020). Furthermore, in order to predict the power production of wind farms, understanding and modelling their interaction with the atmospheric boundary layer is needed. This interaction defines the entrainment of energy into the wakes of the wind turbines (Stevens and Meneveau, 2017).

Despite decades of intensive studies, even the modelling of averaged statistics of velocity and other quantities (such as Reynolds stresses, dissipation, etc...) remains an open question (Johansson et al., 2003; Nedic et al., 2013). For instance, the existence of turbulent wakes with non-canonical energy cascades that result in modifications to the streamwise scaling of the averaged velocity deficit and the wake width has recently been reported (Neunaber et al., 2022c; Ortiz-Tarin et al., 2021). Moreover, several aspects of this flow, particularly relevant for wind energy applications, are still under debate, such as the

influence of the background flow and the characteristics of the testing facility (Aubrun et al., 2019; Biswas and Buxton, 2024; Hearst et al., 2016), the spatial extent of the turbulence production region and the near wake (Neunaber et al., 2024; Vahidi and Porté-Agel, 2022; Gambuzza and Ganapathisubramani, 2023), and how the latter is modified by the operating conditions of the rotor (Bourhis and Buxton, 2024; Neunaber et al., 2022a; Scott et al., 2024).

In this context, in many important situations, turbulent wakes evolve within a pressure gradient (Hill et al., 1963; Liu et al., 2002). For instance, wind plants installed over complex topographies and tidal stream generators are expected to be potentially affected by pressure gradients that modify the structure and persistence of the wakes. Consequently, the problem has received renewed attention in recent years (van der Deijl et al., 2022; Cheng et al., 2024). Predictions for the streamwise evolution of the normalized averaged velocity deficit and the wake width have been proposed for the planar (Shamsoddin and Porté-Agel, 2017) and axisymmetric (Shamsoddin and Porté-Agel, 2018) turbulent wake. Furthermore, these proposed scalings have been verified experimentally and numerically for cylinders, disks, and scaled wind turbines (Dar and Porté-Agel, 2022; Dar et al., 2023; Dar and Porté-Agel, 2024).

In this work, we report a wind tunnel study using porous actuators to further characterize the effect of an adverse pressure gradient (APG) on vertical and horizontal axis wind turbine wakes (HAWTs and VAWTs, respectively). Indeed, porous plates have been found to properly match several averaged properties of HAWT wakes (Aubrun et al., 2013, 2019; Camp and Cal, 2016; Vinnes et al., 2022), and even their higher-order statistics (Vinnes et al., 2023; Neunaber et al., 2022b). On the other hand, VAWTs have also been modelled as porous cylinders (Steiros et al., 2020; Ning, 2016). We remark, nevertheless, that in the present study the cylinder only emulates a VAWT with a very large aspect ratio since the cylinder spans the full height of the wind tunnel. The aspect ratio of a VAWT is important for the development of the wake (Araya et al., 2017), so this cylinder is not a perfect representation of a VAWT wake.

We present a systematic study in which a porous disk and a porous cylinder are characterized within a streamwise zero pressure gradient (ZPG) and an APG. Both generators are tested at three different Reynolds numbers ($Re_D = U_\infty D/\nu$, with $U_\infty$ the freestream velocity, $\nu$ the kinematic viscosity, and $D$ the diameter of the disk or cylinder) in the range $2.6 \times 10^5 - 3.9 \times 10^5$. They are also designed to have similar values of thrust coefficients $C_T$ that match realistic rotors.

To this aim, the wake generator can be placed in the normal test section of the wind tunnel (with a negligible pressure gradient) or in the diffuser section, where a constant APG is present in the flow. The wind tunnel has a long diffuser section, allowing the study of the wake's evolution at least 7 diameters downstream of its generator. It, therefore, covers a range that is pertinent for wind energy applications, particularly concerning turbine layouts within a farm. Furthermore, we consider one case where the generator is placed in the test section, and the wake evolves within it and then continues through the diffuser. Consequently, our work covers both the situation in which a self-similar turbulent wake faces a pressure gradient and the one in which both the near and far wakes evolve within the APG.

The relevance of this study lies in the comparison, at relatively large values of $Re_D$, of the turbulent wakes of porous cylinders and disks. Our results, obtained in the same tunnel can be used and adapted to the design of HAWTs and VAWTs in the presence of pressure gradients. They also allow us to evaluate how different the effect of an APG is for both types of rotors, as experiments are performed in the same facility and using the same collection techniques. Our experimental results are also

compared with available analytical models for turbulent wakes within an APG (discussed in detail in section 2, see also, for instance, (Shamsoddin and Porté-Agel, 2018; Shamsoddin and Porté-Agel, 2017)), finding a satisfactory agreement between them.

The present manuscript is organized as follows. Section 2 summarizes the models from the literature that describe the streamwise evolution of turbulent wakes in the presence of a pressure gradient (Shamsoddin and Porté-Agel, 2017; Shamsoddin and Porté-Agel, 2018). Section 3 details the experimental setup, including the wind tunnel used for the tests, the tested objects, and how velocity profiles were obtained. Section 4 discusses the effect of the APG depending on the type of generator. In section 5, the influence of the pressure gradient in the turbulent wakes is evaluated following the models from the literature. Finally, section 6 states the conclusions and perspectives raised by this work.

## 2  Theory: streamwise evolution of turbulent wakes in an adverse pressure gradient

The theory that models the far turbulent wake always requires self-similarity of the averaged streamwise velocity deficit (Bastankhah and Porté-Agel, 2014). Additionally, further terms from the momentum or kinetic energy balances can be modelled if this property is imposed to other quantities (George, 1989; Townsend, 1976). In the following, we will briefly detail the models available to describe the evolution of a turbulent wake within a ZPG and in the presence of an APG. We remark that, while in the present work only a constant APG is considered, the models detailed below work for any type of pressure gradient (including adverse or favorable conditions).

### 2.1  Axisymmetric wake

In the case of a self-similar axisymmetric wake, the mean velocity deficit can be appropriately described by a Gaussian profile (Pope, 2000), such that,

$$\frac{U_b(x) - \bar{u}(x,r)}{U_b(x)} \equiv C(x) \, e^{-\left(r^2/2\delta^2\right)}, \tag{1}$$

where $U_b$ is the base flow velocity that can vary in the streamwise direction $x$ as a result of a pressure gradient. $\bar{u}$ is the averaged streamwise velocity at the streamwise distance $x$ and a radial distance from the wake centre $r$. $C(x)$ is therefore the centreline (and maximum) velocity deficit in the wake and $\delta(x)$ is the wake width.

Shamsoddin and Porté-Agel (Shamsoddin and Porté-Agel (2018)) derived a nonlinear ordinary differential equation (ODE) from the conservation of the averaged momentum equation for an axisymmetric wake, in which the inviscid terms have been neglected, including a pressure gradient in the x-direction,

$$\frac{dC(x)}{dx} = \frac{-1}{\left(\frac{U_b^4(x)}{\lambda_0^2(x)}\right)\left(3C^2(x) - 2C^3(x)\right)} \left[ \frac{1}{4} \frac{dU_b^4(x)}{dx} \frac{C^3(x)}{\lambda_0^2(x)} + \left(C^3(x) - \frac{C^4(x)}{2}\right) \frac{d}{dx}\left(\frac{U_b^4(x)}{\lambda_0(x)}\right) \right], \tag{2}$$

where $\lambda_0(x)$ is the ratio between velocity deficit and wake width. This ratio $\lambda_0(x)$ is assumed to be independent of the pressure gradient (see equation 4 below). The subscript 0 indicates a quantity in the ZPG. The subscript $i$ refers to the smallest $x$-position

within the pressure gradient and it is where the boundary condition, i.e. the starting point, is applied to the ODE. This boundary condition for this ODE is set to be,

$$C(x_i) = C_0(x_i). \tag{3}$$

It is therefore assumed that the centreline velocity deficit $C(x_i)$ in the case of a pressure gradient is equal to the velocity deficit without pressure gradient, $C_0(x_i)$, at the starting streamwise position of the model. Together with the assumption of self similarity, this means that only a fully developed axisymmetric and self similar wake is exposed to a pressure gradient. And for these assumptions to hold true, the pressure gradient (and the proposed model) cannot start in the near wake. Accordingly, it

requires that the wake generating object is not placed in a pressure gradient, as this would mean that the velocity deficit would not be equal at the starting position of the model. However, Dar and Porté-Agel (2022) propose a correction for the case where the pressure gradient starts in the near-wake or the object is placed in the pressure gradient, which will be discussed below in section 2.2.

The final assumption that is made by Shamsoddin and Porté-Agel (2018) to deduce the ODE of equation 2, is that the ratio

of the maximum velocity deficit and wake width is unaffected by a pressure gradient. This allows to apply the ratio of the ZPG case to the model:

$$\lambda(x) = \lambda_0(x) = \frac{U_{b0}C_0}{\delta_0}, \tag{4}$$

where the subscript $0$ always denotes the ZPG case. This assumption of the invariance to the pressure gradient was proven by Liu et al. (2002)) and Thomas and Liu (2004).

With equations 3 and 4, the ODE of equation 2 can be solved and with this the velocity deficit and wake width of an axisymmetric self-similar wake in a pressure gradient can be estimated. The only requirement is to know the values of the centreline velocity deficit $C_0$ and the wake width $\delta_0$ of a ZPG wake and the base flow velocity within the pressure gradient $U_b(x)$.

For a ZPG axisymmetric wake, the model is consistent with the results from Bastankhah and Porté-Agel (2014), that imply,

$$C_0(x) = \left(1 - \sqrt{1 - \frac{c_T}{8\left(\frac{k_{BP}x}{D} + 0.2\sqrt{\gamma}\right)^2}}\right), \tag{5}$$

with $\gamma = \frac{1+\sqrt{1-c_T}}{2\sqrt{1-c_T}}$. $k_{BP}$ is the growth rate of the wake and depends on mostly on the atmospheric turbulence intensity. Nevertheless, the streamwise functional forms are only necessary to give a base condition for the ZPG in the boundary condition of equation 3 (this point is also further discussed in section 5). For that reason, instead of equation 5 we have used the power-law fits derived from the Townsend-George theory (Townsend, 1976; George, 1989), that give a joint prediction for the normalized

velocity deficit and the wake width $\delta(x)$:

$$C_0(x) = A\left(x - x_0\right)^{-\alpha}, \tag{6}$$

$$\delta(x) = B(x - x_0)^\beta. \tag{7}$$

The constants $A$, $B$, $\alpha$ and $\beta$ can be related via momentum conservation, but in this work they are considered as independent fitting parameters. The virtual origin $x_0$, also a tunable quantity, is identical in both equations. Indeed, such 5-parameter fits are a standard procedure in experimental studies on turbulent wakes as the discretised nature of the streamwise positions do not usually allow to perform a reliable 3-parameter fit (Nedic et al., 2013). Moreover, the constants are expected to depend not only on the generator but also on the background turbulence intensity and the tip speed ratio (Bourhis and Buxton, 2024; Neunaber et al., 2022a). In consequence, for each experimental conditions the values of $C_0(x)$ and $\delta(x)$ can be extracted, and adjusted simultaneously using equations 6 & 7. The resulting streamwise scalings are then used to feed the ODE from equation 2 via the boundary condition from equation 3. As it will be discussed in the next section, this approach has also the advantage that the power-law fits can be used for a cylinder-generated wake, allowing to set the boundary conditions of ODEs using a common protocol.

## 2.2 Wake generator placed within the pressure gradient

As discussed, the ODE that was shown above (equation 2) assumed that the object that generates the wake does not experience a pressure gradient and only a developed, self-similar wake develops within it. In consequence, the pressure gradient only affects the far (in the sense of a self similar) wake. Nevertheless, in many potential applications the pressure gradient does not start in the far wake. Dar and Porté-Agel (2022) propose a modification to the boundary condition of equation 3 to correct for the change in the base velocity $U_b$ due to a pressure gradient. In this specific case, the starting condition is different because the wake generator and the near wake are both immersed in the pressure gradient. Therefore, they propose that equation 3 becomes,

$$C(x_i) = 1 - \frac{U_{nw}(x_i)}{U_b(x_i)}. \tag{8}$$

Because the velocity profiles in the near wake are not Gaussian, the velocity in the centre of the wake in the near wake, called $U_{nw}$, is used to adjust the velocity deficit $C(x_i)$ in the boundary condition of the ODE. This correction has been validated using data from scaled rotor wakes in a wind tunnel. However, instead of a derivation from the ZPG case, in this study the starting point of the ODE is simply chosen to be the first data point of the APG where the wake is self-similar. This starting point is estimated to be downstream $x/D \sim 3$, as will be discussed in section 4.1. The information on the velocity deficit at this starting point is available for our case and it allows for a better prediction by the model across the whole wake within the APG.

## 2.3 Planar wake

In addition to an axisymmetric wake, a similar solution was derived by Shamsoddin and Porté-Agel (2017) for two-dimensional planar wakes. This solution has a similar form as equation 2 and is again an ODE:

$$\frac{\mathrm{d}C(x)}{\mathrm{d}x} = \frac{-1}{\left(\frac{U_b^3(x)}{\lambda_0(x)}\right)\left(2\sqrt{2}C(x) - 3C^2(x)\right)}\left[\frac{\sqrt{2}}{3}\frac{\mathrm{d}U_b^3(x)}{\mathrm{d}x}\frac{C^2(x)}{\lambda_0(x)} + \left(\sqrt{2}C^2(x) - C^3(x)\right)\frac{\mathrm{d}}{\mathrm{d}x}\left(\frac{U_b^3(x)}{\lambda_0(x)}\right)\right]. \tag{9}$$

Aside from the final ODE, the approach and assumptions are the same as described in the previous sections 2.1 and 2.2, including an identical boundary condition as the one stated in equation 3. Moreover, the functional forms for the scalings of $C(x)$ and $\delta(x)$ are identical between an axisymmetric and a planar wake, as they both can be adjusted with power laws (Townsend, 1976; George, 1989), the main difference between both flows concerns the values of the exponents $\alpha$ and $\beta$. In consequence, the boundary conditions for the planar wake will also be taken using fits following equations 6 & 7. In section 5, we will discuss how these theoretical models and assumptions compare to the experimental results.

## 3 Experimental setup

The experiments were performed in the subsonic S620 wind tunnel of ISAE-ENSMA in Poitiers (figure 1). It has a 6 m long test section with a cross section with a width $W$ of 2.4 m and a heigh $H$ of 2.6 m. The wake generator can be installed in the test section or within the diffusing section of the wind tunnel. The latter remains accessible and spans for 10 meters, beyond the range covered by this work (that is up to approximately 4.1 m downstream this section when a generator is present). Within that range, it has a constant expansion angle in the four walls of $3°$. The cross sectional area of the tunnel in that range is therefore given by,

$$A(x) = (H + 2x\tan(\gamma))(W + 2x\tan(\gamma)), \tag{10}$$

with $x$, in this case, the streamwise distance from the start of the diffuser section (and not to the generator as in the other cases) and $\gamma$ the angle of the walls. The expected baseline velocity $U_b$ at $x$ is then related to the measured velocity in the test section $U_\infty$ as,

$$\frac{U_b(x)}{U_\infty} = \frac{A(0)}{A(x)}. \tag{11}$$

The freestream velocity $U_\infty$ at the inlet was measured above the turbine at the ceiling of the tunnel. The turbulence intensity for an empty test section, defined as the ratio between the standard deviation of the streamwise velocity and its averaged value, is of 0.25%. This parameter was measured in a previous work using hot-wire anemometry (Myskiw et al., 2024) and is given as an indicator of the base flow quality for reproducibility purposes.

Fifteen Pitot tubes were positioned on a horizontal rack, with a single static pressure probe providing the static pressure. This static pressure probe was positioned in the centre of the rack and slightly above of it. The 16 pressure channels were calibrated and recorded with a DTC Initium pressure scanning system at a frequency of 1 kHz. The resolution in terms of pressure of

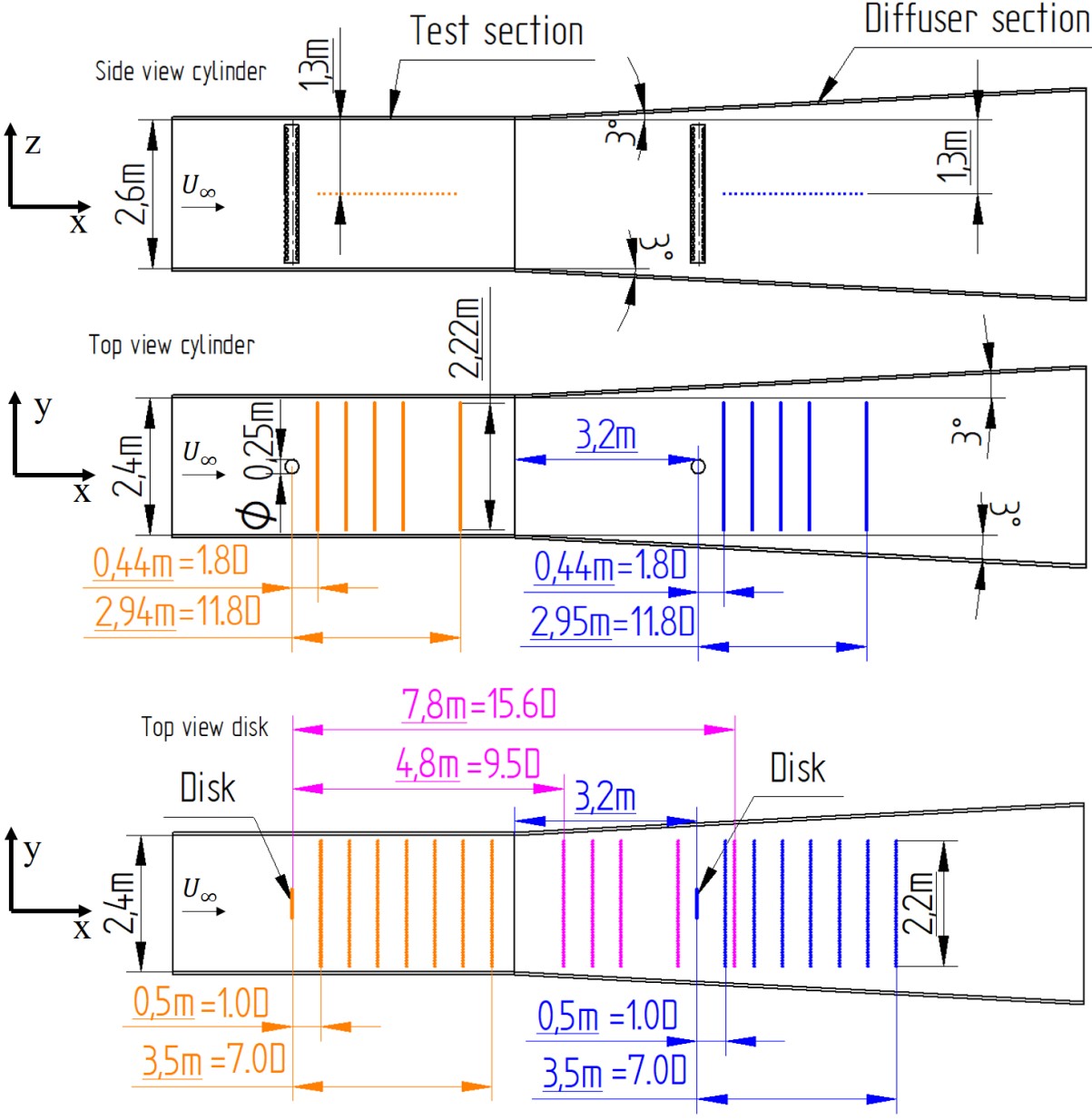

**Figure 1.** Sketch of the side view of the experimental setup (upper panel) and the top view when the cylinder (middle panel) and the disk (lower panel) are installed. The test section has a length $L_{TS}$ of 6 m. The wake generators positions, were either at the inlet of the test section or at $L_{DS} = 3.2$ m, downstream the beginning of the diffuser section. A rake of pressure tubes allowed to do profiles in the central plane of the generator, exploring the $x - y$ plane.

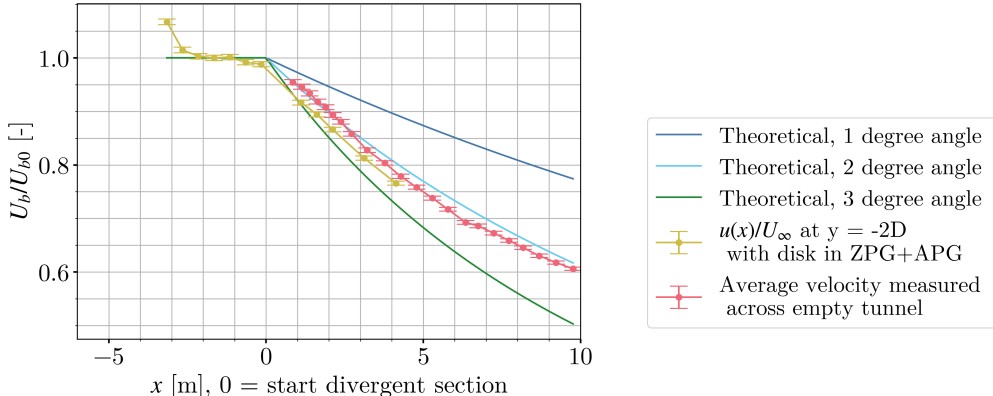

**Figure 2.** Normalized baseline velocity within the test and diffuser sections for an empty wind tunnel (orange line) and for the disk at $y = -2D$ (violet line). The latter is a lateral position that lies outside of the wake throughout all the experimental conditions tested. Figures are compared to the variation of velocity expected from different expansion angles of the walls, deduced via flow rate conservation.

the system is of 0.05%. This implies an absolute error of 0.025% for the smallest velocity recorded. Nevertheless, given other sources of error in the velocity measurement (such as the calibration of the pressure system), we estimate an absolute error of velocity measurements at 1%. The minimum duration of each measurement was 60 s, allowing to resolve and converge the average and rms values of the velocity. As only the averaged and standard deviation values of the signal were extracted, no filter was applied to the raw data. In some cases, very near the generator, recorded pressure was negative due to the flow reversal in the recirculation region downstream the disk and the cylinder. In such cases (see for instance figure 5a below), the recorded velocity was not taken into account in calculations.

The spatial separation of these fifteen pressure tubes on the rack was fixed at 15 cm, but the final resolution was increased by off-setting the rack transversely in successive measurements. This meant that for each downstream position 3x15 positions with 5 cm spacing for the disk and 5x15 positions with a 3cm resolution for the cylinder were recorded. The measurements also span up to 10 cm and 9 cm close to both side walls (in the normal section) for the disk and the cylinder, respectively. In the diffuser section the closest distance to the walls recorded ranges from 29 cm to 47 cm for the disk and from 28 cm to 44 cm for the cylinder. The error in the position is estimated as 2 mm. On overview of all measurement station can be seen in table 1.

| Section | Disk | | | Cylinder | | |
|---|---|---|---|---|---|---|
| | TS | DS | TS+DS | TS | DS | TS+DS |
| Streamwise range ($x/D$) | 1, 2, 3, 4, 5, 6, 7 | 1, 2, 3, 4, 5, 6, 7 | 9.5,10.5, 11.5, 13.5, 15.6 | 1.8, 3.8, 5.8, 7.8, 11.8 | 1.8, 3.8, 5.8, 7.8,11.8 | × |
| Lateral increment ($\Delta y/D$) | 0.1D | 0.1D | 0.1D | 0.12D | 0.12D | × |

**Table 1.** Overview of the streamwise distances measured with the rake of pressure tubes, including tests performed for the cylinder and the disk in the test section (TS), diffuser section (DS) and jointly if both sections (TS+DS). The sign × implies that no tests were carried on for those conditions. The lateral increment (i.e. the spatial resolution of acquired lateral profiles) is also given.

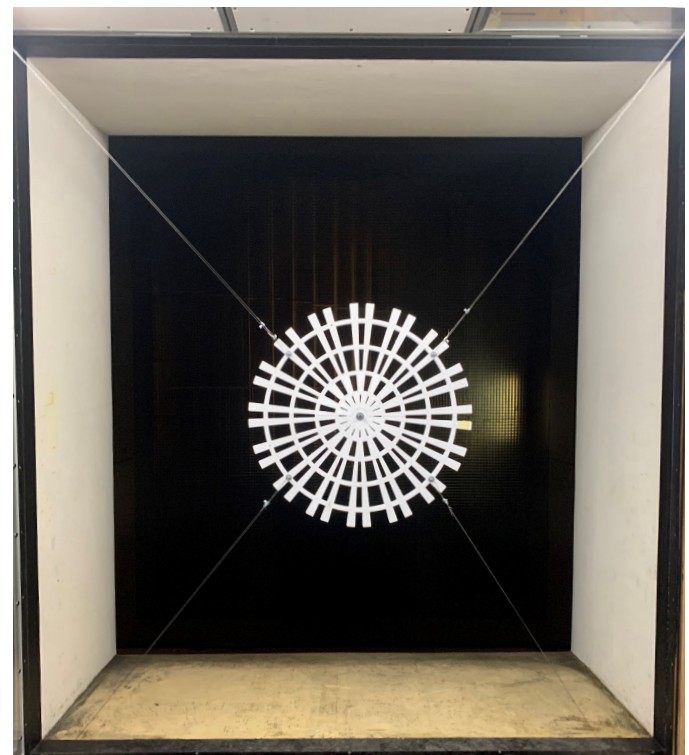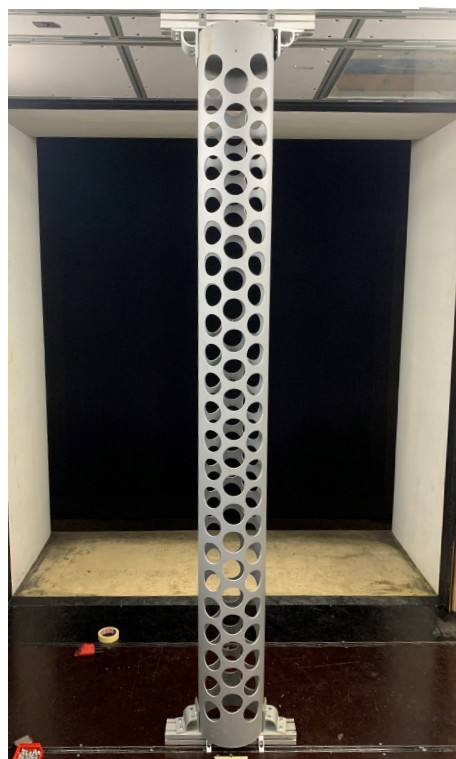

**Figure 3.** Porous disk (left) and porous cylinder (right) installed at the inlet of the test section, as used in part of this study. The diameters of the disk and the cylinder are of 0.5 m and 0.25 m, respectively.

Figure 2 shows the evolution of the averaged streamwise velocity for an empty test section and the evolution of the velocity measured outside the wake. It is observed that the velocity within the diffuser is consistent with a $2°$ uniform expansion, slightly different to the geometrical expansion of the walls, as governed by equations 10 and equation 11. This may be due to the development of boundary layers at the walls, and we will therefore consider that the APG in our flow is caused by the $2°$ we observe experimentally. When the disk is present, a small acceleration is also appreciated very near the generator, caused by the blockage effect of it.

Two wake generators were tested, a porous circular cylinder and a porous plate (see figure 3), designed in a way that they had matching drag coefficients. All measurements were made for three different freestream velocities (and therefore different $Re_D$), and profiles cover several streamwise distances. As stated, they were also made placing the generator either in the test or in the diffusing section. For the latter, the generator was placed 3.2 m downstream after the beginning of the diffuser. For the former, it was placed almost at the inlet of the test section. In the following, we will detail the set of measurements performed for each wake generator.

## 3.1 Porous disk

The disk is made according to the design proposed by Neunaber et al. (2021), that presents a diminishing blockage with the distance from the plate's center. It has been manufactured in plastic and has a diameter of $D = 0.5$ m, that results in a blockage of 3%. It was held at the centre of the section using 0.75 mm piano wires. The thrust coefficient of the plate is $C_T \approx 0.9$ and the porosity 47% (the coefficient $C_T$ was not measured here, and the value reported by Neunaber et al. (2021) was used instead).

Measurements include three different freestream velocities, $U_\infty = 7.8$ m/s, 9.2 m/s and 11.7 m/s, that correspond to values of Reynolds numbers of $Re_D = 2.6 \times 10^5$, $3.1 \times 10^5$ and $3.9 \times 10^5$, respectively. Horizontal profiles (i.e. in the $y$ direction), for the disk placed either in the test or in the diffuser section, were taken at the same distances with respect to it, being approximately: $x/D = 1, 2, 3, 4, 5, 6$ and $7$ (see table 1 for the exact values).

For the porous disk an extra profile was measured, where the plate was placed at the entrance of the test section and measurements were done at the diffuser, adding therefore the streamwise distances (all within the diffuser section) of $x/D = 9.5, 10.5, 11.5, 13.5, 15.6$.

## 3.2 Porous cylinder

The cylinder has a diameter $D = 0.25$ m and spans vertically the whole section and is fixated to the floor and the ceiling. It is made of PVC and has 136 circular holes with a diameter of 74 mm, resulting in a porosity, relative to the frontal area, of 43.5%. It was made following a previous study design (Steiros et al., 2020), that reports a thrust coefficient $C_T \approx 0.9$ (matching the value of the disk). The total blockage of the cylinder is of 6%.

The generator was tested at three different streamwise velocities $U_\infty =15.6$ m/s, 18.4 m/s and 23.5 m/s, that correspond to values of $Re_D$ that match the ones for the disk. Horizontal profiles in the test section were taken at approximately $x/D = 2, 4, 6 ,8$ and $12$ (see table 1 for the exact values). When the plate was placed at the diffuser section, the same streamwise distances were recorded.

## 4 Results

In this section we will discuss the effect of the pressure gradient on the velocity deficit and the wake width for both tested objects, as well as the self similarity of the wake and the effect of Reynolds number. First, we will discuss the raw statistics deduced from our measurements and we will show that, in the conditions studied here, the flow can be considered self similar (section 4.1). Later, in section 4.2, will discuss the validity of the assumption of the invariability of $\lambda_0$ to the pressure gradient. In section 4.3, we will focus in the differences in terms of velocity deficit and wake width between both generators and in the presence or not of an APG. Moreover, we will assess if the turbulent wakes still present Reynolds number effects (section 4.4). This analysis will be further used in section 5 to apply and discuss the models available in the literature to quantify the effect of an APG.

## 4.1 Velocity deficit, wake width and self similarity

We start this section by discussing the averaged velocity profiles obtained for all conditions tested. Figure 4 shows the normalized averaged streamwise velocity $u/U_\infty$ obtained for different values of $x$ and $y$. From this velocity contour plot (figure 4) it is already clear that the wake evolves differently when it is subject to an APG; in the presence of the APG, both the velocity deficit and the wake width increase.

While this difference is obvious when normalizing with the constant $U_\infty$, it is also clear that the wake evolves differently if one normalizes with the velocity $U_b(x)$ (that is a decreasing function for icnreasing $x$), as shown in the velocity profiles of figures 5a and 5b. Relatively to the ZPG case, the velocity deficit and wake width are both increased.

This can also be seen in figures 6a and 6b, which show the evolution of the velocity deficit and wake width, respectively, of the wakes behind the disk and cylinder at $Re_D = 3.9 \times 10^5$. The velocity deficit seems to evolve with the same slope, regardless of the pressure gradient. Nevertheless, this cannot be said about the wake width, which appears to be strongly affected by the APG. This is not an unexpected behaviour, as the velocity deficit and wake width are related; also shown through equation 4: $\lambda(x) \sim \lambda_0(x)$ for the self similar streamwise range ($x/D \geq 3$, see discussion below). We have verified that regardless of the pressure gradient, the ratio of velocity deficit and wake width is the same for both the ZPG and APG, having only a dependency in $x$.

Furthermore, for both the disk and cylinder, it can be appreciated that the profiles $\frac{U_b(x) - \bar{u}(x,r)}{\Delta u(x)}$ collapse when the radial distance is normalized with the wake width $\delta(x)$ (figures 5c and 5d). In all cases, $\delta$ is estimated as the standard deviation of a Gaussian fit applied to the velocity profile $U_b(x) - \bar{u}(x,r)$ at a given $x$-position. $\Delta u(x)$ is the maximum velocity deficit at a given $x$-position. It is defined as $\Delta u(x) = U_b(x) - min(\bar{u}(x,y))$. In consequence, we can conclude that the wake, for all cases studied, in the range $x/D \geq 3$, can be considered as self similar (at least in terms of the averaged velocity field).

Finally, it must be noted that the wake of the cylinder is slightly skewed. Our measurements for an empty test and diffuser section show that the baseline velocity in the tunnel is not asymmetric. Therefore, this effect is most likely due to the cylinder positioning. While great care was taken in positioning the cylinder in the tunnel, it is possible that a very small angle was created between the centreline of the holes and the incoming flow, causing a small skewness on the wake.

## 4.2 Invariability of $\lambda$ to pressure gradient

As mentioned above, the ratio of velocity deficit to wake width, as given by equation 4, is a key relation in the model proposed by Shamsoddin and Porté-Agel (2018). In the studies by Liu et al. (2002) and Thomas and Liu (2004), it is shown that $\lambda$ is invariant to the pressure gradient. This invariance is also depicted in figure 7 for the disk and cylinder at $Re_D = 3.9 \times 10^5$. From this figure, it can be observed that the evolution of this parameter in the streamwise direction in the wake of either wake generator follows a very similar pattern, regardless of the pressure gradient. Therefore, using this invariability to solve the ODEs in equations 2 and 9 appears to be a valid assumption.

However, there is an offset in the absolute value between the ZPG and APG cases. This offset is not shown in the studies by Liu et al. (2002) and Thomas and Liu (2004), as their work scales the value of $\lambda(x)$ relative to $\lambda(0)$ at $x = 0$. A similar

scaling has been applied to figure 7a, shown in figure, 7b, such that $\lambda(x)$ is scaled to the first measured value. In the case of the cylinder the two curves collapse. In the case of the disk, there is a small absolute offset between the ZPG and APG case. In the far wake the curves seem to reach the same asymptote. Scaling it to the first value does not seem to work well, because the near-wake is difficult to characterize due to the negative velocities encountered close to the disk. Thomas and Liu Thomas and Liu (2004) note that differences in absolute value may arise due to differences in the Reynolds numbers. It appears that small differences between the experimental setups for the ZPG and APG cases have resulted in this small difference in the absolute value of $\lambda(x)$. Unfortunately, this offset remains relatively constant, causing a relative difference of 20%-30% between the two cases at larger values of $x/D$. As will be shown in section 5, variations in $\lambda(x)$ result in only minor errors in the calculated velocity deficit. However, since the calculation of wake width in the model by Shamsoddin and Porté-Agel (2018) is directly proportional to the absolute value of $\lambda_0$, any uncertainty in this parameter directly translates to the same level of uncertainty in the estimated wake width. This means that the 20%-30% relative difference is also present in the wake width. Thus, while the invariability of $\lambda(x)$ to pressure gradient is confirmed, it is critical to measure its absolute value accurately, especially in the near-wake.

### 4.3 Differences between disk and cylinder

Remarkably, the effect of the APG is significantly different for the two different generators. For instance for the disk the value of $1 - C(x)$ is approximately 7-9% smaller for the APG with respect to the ZPG, while the wake width $\delta$ is larger, with a difference ranging from 21% at $x = 3D$ and 47% at $x = 7D$ (figures 6a and 6b). On the other hand, the differences for the cylinder are of 16-20% for $1 - C(x)$ and range from 10% to 20% for $\delta$.

From the velocity profiles in figures 5a and 5b, it can be observed that for the disk, very near the generator ($x = 1D$ to $x = 2D$), the velocity becomes negative. The profiles are incomplete, as the Pitot tubes cannot measure these negative velocities. This is not the case for the cylinder: even if the $C_T$ of both objects was designed to be the same, the near wakes generated by these two objects differ significantly. This can be attributed to the fact that the wake of the cylinder is two-dimensional, while the disk has a three-dimensional wake. It is not unexpected that a two dimensional case has a stronger wake.

Finally, the velocity profiles in the wake of the cylinder do show two other significant differences between the ZPG and APG case. First of all at $x = 2D$, the wake in the APG is not yet Gaussian. It appears it takes a slightly longer distance for the velocity profiles, as shown in figure 5d, to collapse as compared to the disk. Second of all, the wake of the cylinder in the APG case is slightly skewed towards negative values of $y$.

### 4.4 Effect of Reynolds number

Figures 8a-d show the streamwise evolution of the centreline velocity deficit (displayed as $1 - C(x)$) and $\delta(x)$ for the three values of $Re_D$ considered in this work. First, the effect of the APG is the same for all Reynolds numbers and generators, increasing the velocity deficit $C(x)$ and increasing the wake width $\delta(x)$. Furthermore, it can be observed that, for the disk, all curves collapse (figures 8a&b) onto a single one, showing a low sensitivity to Reynolds number. The cylinder, displayed

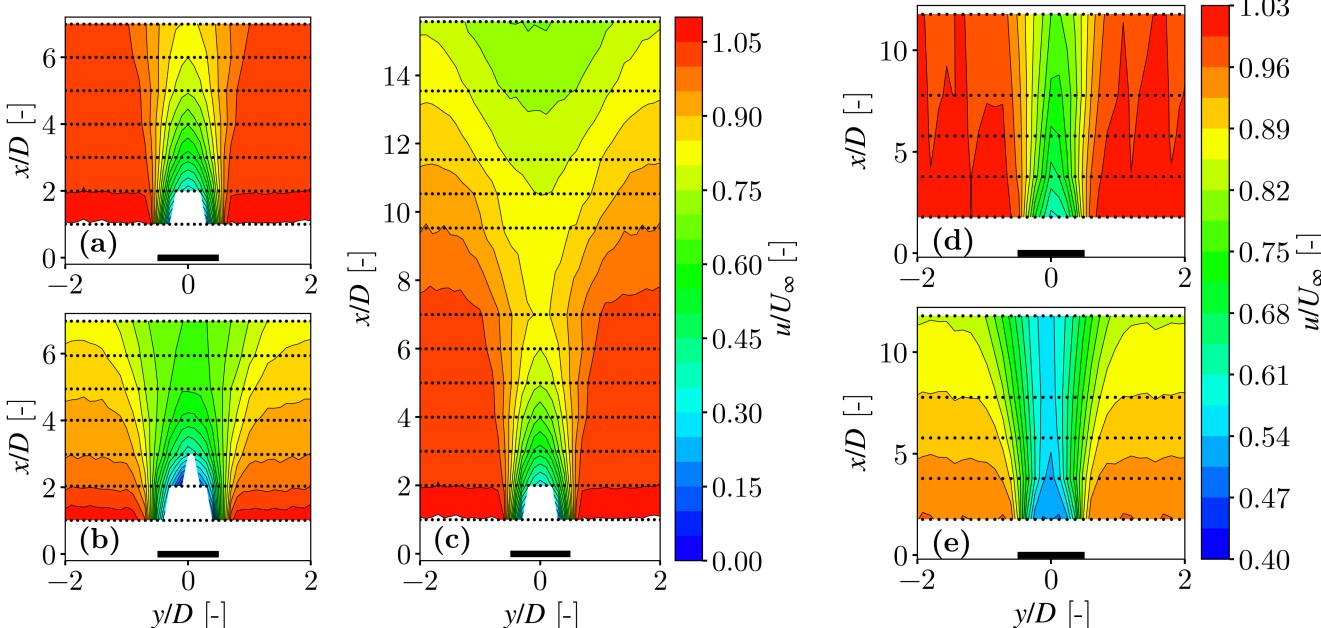

**Figure 4.** Horizontal profiles of the normalized averaged streamwise velocity $\bar{u}/U_\infty$ obtained for all cases studies: disk placed in the test section (a), in the diffuser section (b) and placed in the test section but expanding through this section and the diffuser one (c). The figure also includes the cylinder placed in the test (d) and in the diffuser (e) section. The black dots represents the points where measurements were taken and the solid black lines the velocity contours. Data was interpolated to generate a map where, in order to better show the details of each wake, the color scale among panels is not identical.

in figure 8c-d shows a similar behaviour for the two larger $Re_D$. However, the smallest Reynolds number ($Re_D = 2.6 \times 10^5$) presents some deviations with respect to the other curves. Identical trends are observed when checking the Reynolds number dependence for the horizontal profiles of velocity deficit (as the ones displayed in figure 5, not shown here for the other values of $Re_D$). We therefore conclude that for all generators, results become independent of $Re_D$ for the two largest values tested.

In the following, we will discuss our dataset in terms of the models introduced in section 2. Given the discussion above, we will report only results collected at the largest value of Reynolds number, $Re_D = 3.9 \times 10^5$.

## 5 Comparison with the models from the literature

We will now focus on how our dataset can be described by the models discussed in section 2. First, to be able to apply the theory, the turbulent wakes have to be self similar, an aspect that has already been verified in the last section.

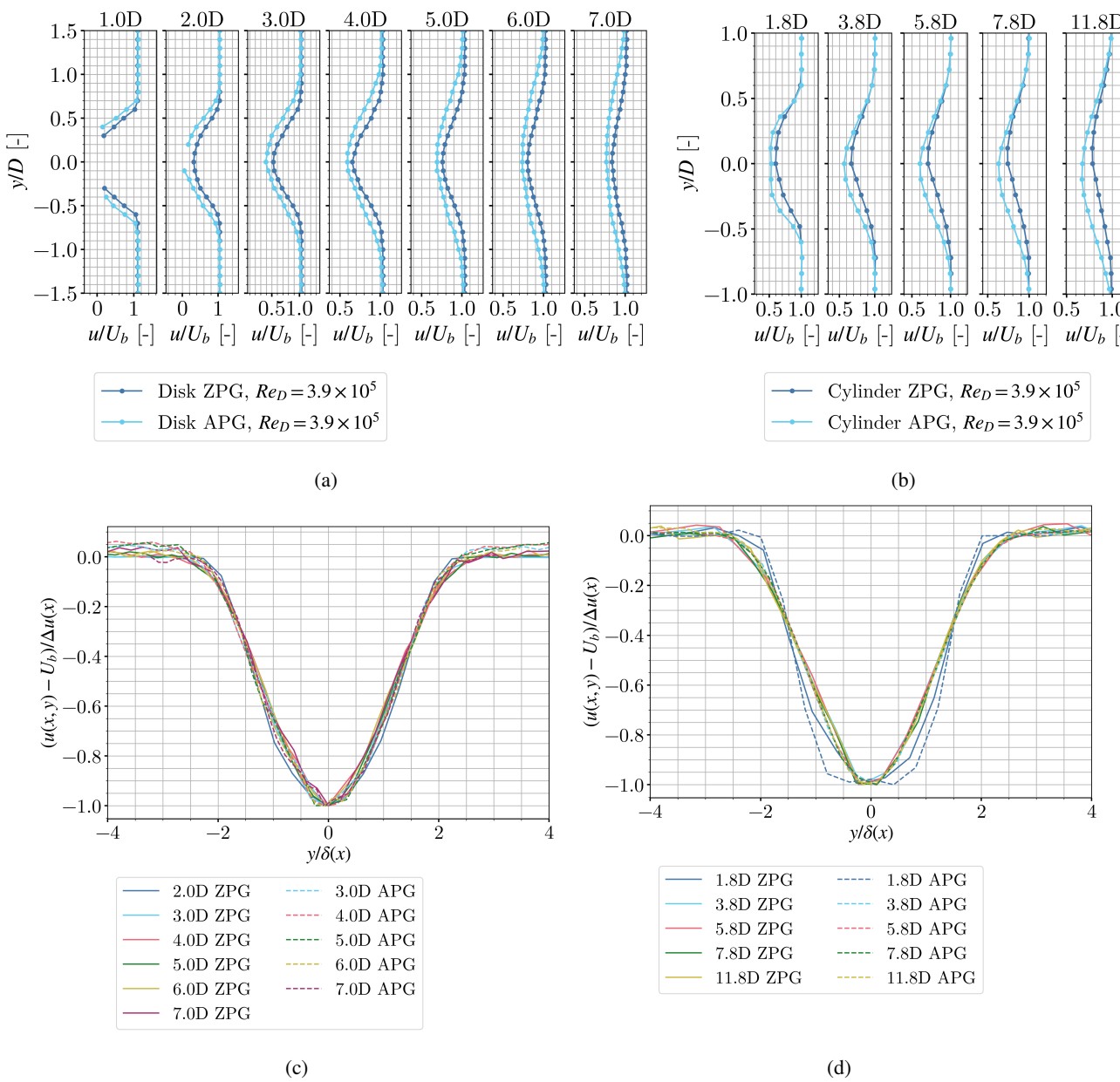

**Figure 5.** Horizontal velocity profiles of $\bar{u}/U_b$ in the wake of the disk (a)) and the cylinder (b)) at different streamwise positions $x/D$ downstream the generators. Same profiles but normalized using the centreline velocity deficit $\Delta u(x)$ and the wake width $\delta$ ((c) and (d) for the disk and the cylinder, respectively). All figures correspond to $Re_D \sim 3.9 \cdot 10^5$. For panels a and b, the error bars are smaller than the marker size. Panels c and d aim at showing a qualitative collapse and therefore no error bars are added.

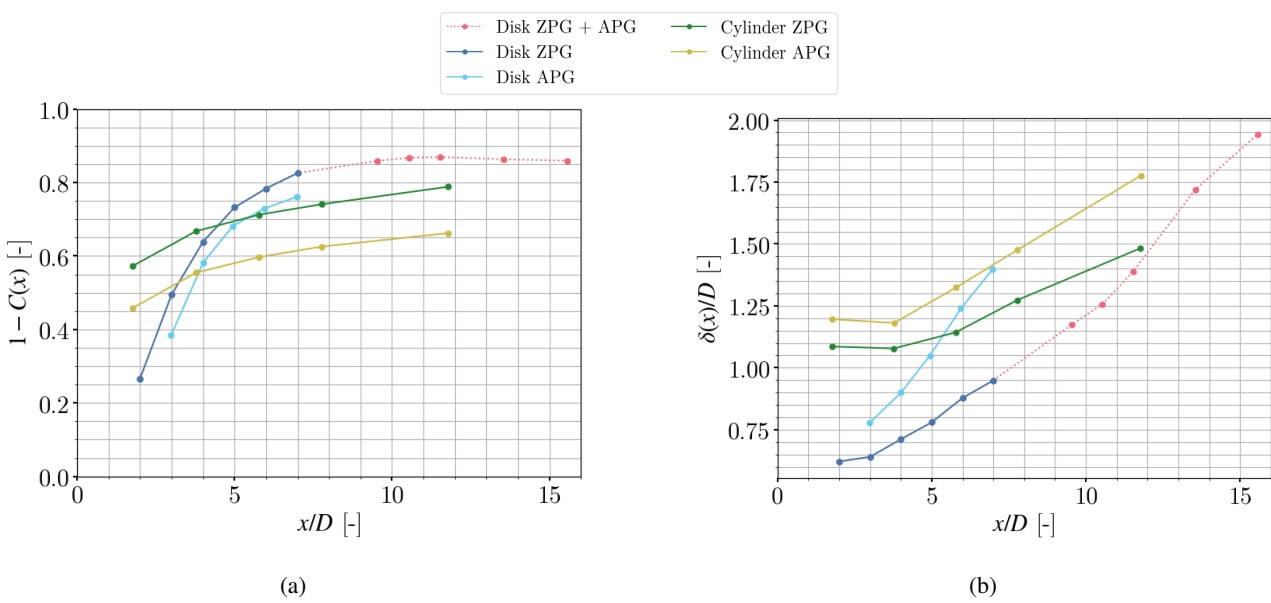

**Figure 6.** Velocity deficit (displayed as $1 - C(x)$) (a) and wake width $\delta(x)$ (b) for all generators and pressure gradients at $Re_D = 3.9 \times 10^5$.

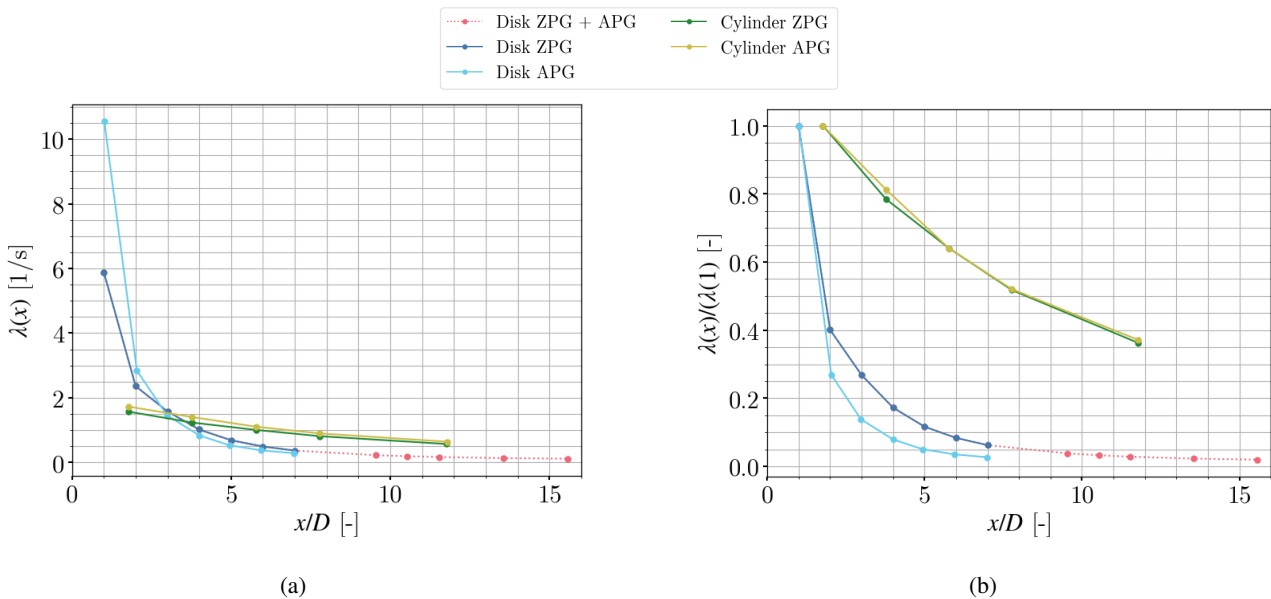

**Figure 7.** Streamwise dependence of the ratio $\lambda(x) = U_b(x)C(x)/\delta(x)$ (equation 4) for all generators at $Re_D = 3.9 \times 10^5$. Left figure (a) shows the absolute value while the right figure (b) normalizes it to $\lambda(x)$ at $x = 1D$.

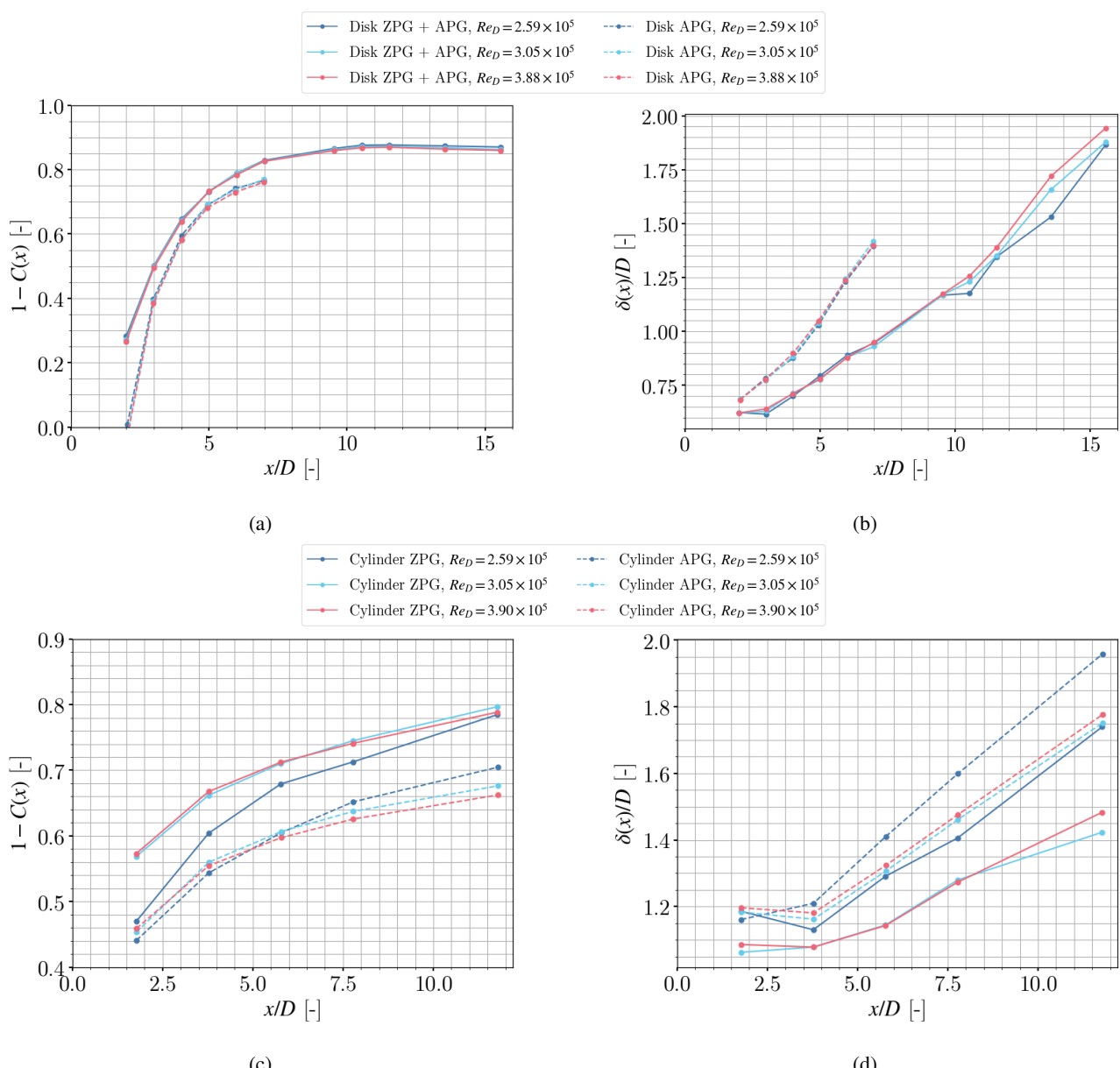

**Figure 8.** Velocity deficit (displayed as $1 - C(x)$) and wake width $\delta(x)$. Influence of the Reynolds number $Re_D$ on these parameters for the disk (figures (a) and (b)) and the cylinder ((c) and (d)).

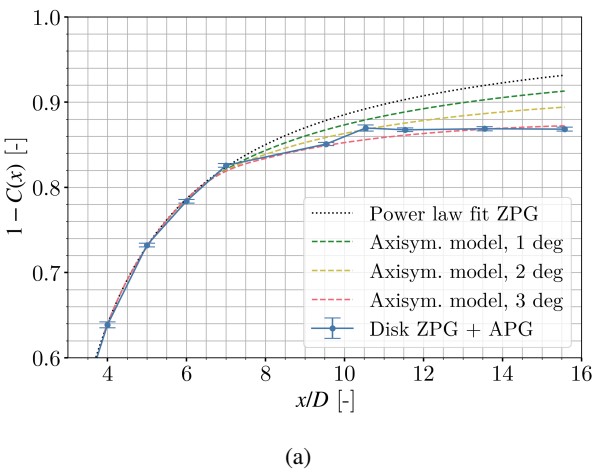 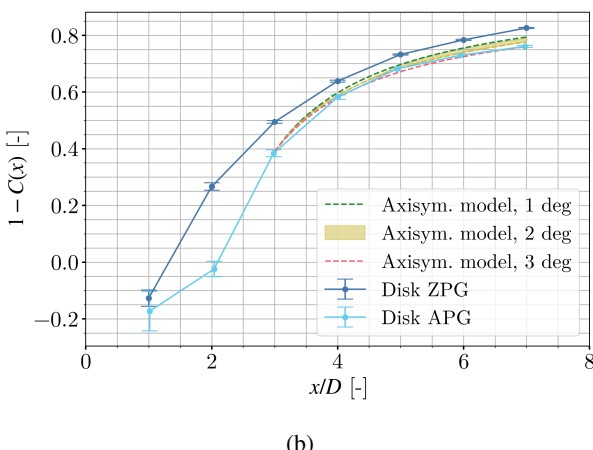

(a)  (b)

**Figure 9.** (a): Velocity deficit $1 - C(x)$ in the wake of the disk with a ZPG continuing into an APG. A power law fit shows how the wake is expected to evolve if there had been no pressure gradient after $x = 7D$. (b): ZPG experimental results versus APG experimental results of the wake behind the disk. Model output with input from the ZPG case is shown for three different wall angles. The shaded area shows the difference between using $\lambda_0$ or $\lambda_{APG}$ as input.

### 5.1 Porous disk

Figures 9b and 9b show the results of the model developed by Shamsoddin and Porté-Agel (2018), described in section 2.1. In particular, figure 9a shows the model applied to a wake starting in a ZPG that continues into an APG. In order to apply the model, a baseline wake in the ZPG case is needed for the entire range up to $x = 15.8D$ as input for the model. Because the length of the test section was limited, no measurements exist after $x = 7D$ for the ZPG case. As discussed in section 2.1, to generate the ZPG velocity deficit and wake width for $x > 7D$, a power-law fit was made according to equations 6 & 7. This fit is based on the ZPG case up to $x = 7D$ and extrapolated until $x = 15.8D$. The velocity deficit and wake width are simultaneously fitted to the two dimensional velocity field as a two dimensional Gaussian fits with a fixed virtual origin. In this Gaussian fit (see also equation 1), the velocity deficit and wake simultaneous fits resulted in the constants: $A = 1.330$, $\alpha = 1.10$, $B = 0.187$, $\beta = 0.51$ and $x_0 = 0.724$. $C_0(x)$ and $\delta_0(x)$ are then used to calculate $\lambda_0$ in equation 4. Then, equation 2 can be solved, obtaining the streamwise scaling of $C(x)$.

The model fits remarkably well both cases considered in our experimental setup: first of all for the APG following a ZPG (i.e., the disk placed at the inlet of the test section and the turbulent wake evolving freely across the test section and into the diffuser, figure 9a). It also models the turbulent wake fully immersed in the APG (i.e. the generator installed in the diffuser, figure 9b). Moreover, while both possible expansions of the diffuser section work (i.e. 2 or 3 degrees), the best fit is found for 3 degrees. This is not fully consistent with the effective expansion found in figure 2 for an empty test section, yet it fits with the actual geometric expansion of the wind tunnel. Nevertheless, by only giving a starting point of $C(x_i)$ and the ZPG wake, the model is able to predict the evolution of the velocity deficit in the wake in an APG extremely well. Small differences in

terms of the expansion angle that fits the generators may also be related to blockage effects and differences in the boundary layer development at the walls in the presence of the wake generators.

The main difference between both situations considered is that when the wake fully develops in the test section, the model can only be applied downstream the near wake, that in our case is $x/D \geq 3$. On the other hand, the model by Shamsoddin and Porté-Agel (2018) properly match the transition of the wake from the ZPG to the APG. This is a reasonable expectation, as the model has been developed to describe the far wake behaviour, where the transverse profiles are Gaussian (or near Gaussian), and the wake is self similar.

Finally, in the case of the disk, the model is quite sensitive to the input $\lambda_0$. This becomes clear from figure 11a, where large differences are observed between the estimated wake widths. Generally, the model is unable to predict the actual wake width, unless the $\lambda_{APG}$ is used. $\lambda_0$ and $\lambda_{APG}$ should be the same, but a small difference, as observed in figure 7 results in a large difference in the wake width.

## 5.2 Porous cylinder

Comparing the evolution of the velocity deficit and wake width in the wakes of the cylinder and the disk in figure 6, one can see that velocity deficit and wake width evolve differently for the wakes of the disk and cylinder. This supports the requirement of using the different ODE of equation 9 for the cylinder. Other than this feature, the porous cylinder presents a similar behaviour as the one described for the axisymmetric wake in the previous section.

As shown in figure 10, the streamwise evolution of the wake in the APG is very well modelled by equation 9. Nevertheless, if this behaviour still holds further downstream ($x > 12D$) remains an open question. In this case, an effective expansion between 1 or 2 degrees works well as input to the model. Furthermore, unlike the disk, the wake width is predicted well with the assumption of $\lambda_0$, as shown in figure 11b. The fits were still performed according equations 6 & 7, resulting in $A = 0.71$, $\alpha = 0.47$, $B = 0.15$, $\beta = 0.35$ and $x_0 = -1.01$. We remark that unlike for the disk, given the limitations of the experimental setup, for the cylinder the case where the cylinder is placed in the ZPG and the wake evolves from the ZPG into an APG has not been considered.

To conclude this section, an excellent agreement between our experimental dataset and the analytical models by Shamsoddin and Porté-Agel (2017) and Shamsoddin and Porté-Agel (2018) has been found for all experimental conditions tested. While our experimental setup is limited in terms of temporal resolution, further studies using hot-wire anemometry and/or particle image velocimetry could help to evaluate further statistics within an adverse pressure gradient. For instance, they would allow to validate some hypotheses from the models, particularly regarding the self similarity and axisymmetry of all relevant terms of the kinetic energy and momentum budgets.

## 6 Conclusions

In this work, we developed an experimental setup specially adapted to assess the influence of an adverse pressure gradient in a wind tunnel. Using the properties of the ISAE-ENSMA wind tunnel in Poitiers, the spatial evolution of a turbulent wake

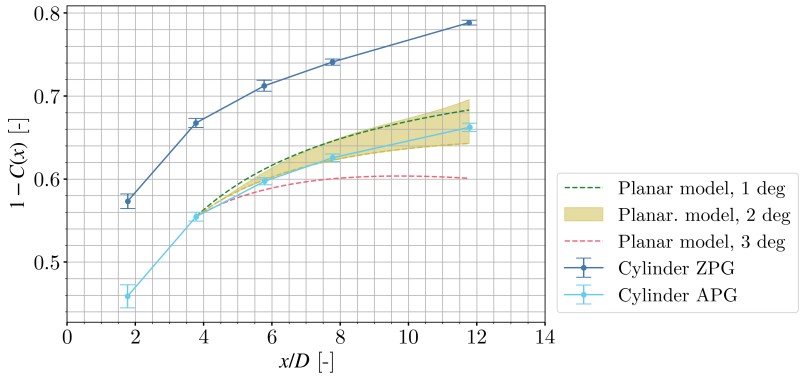

**Figure 10.** ZPG experimental results versus APG experimental results of wake behind the cylinder. Model output with input from the ZPG case is shown for two different wall angles. The shaded area shows the difference between using $\lambda_0$ or $\lambda_{APG}$ as input.

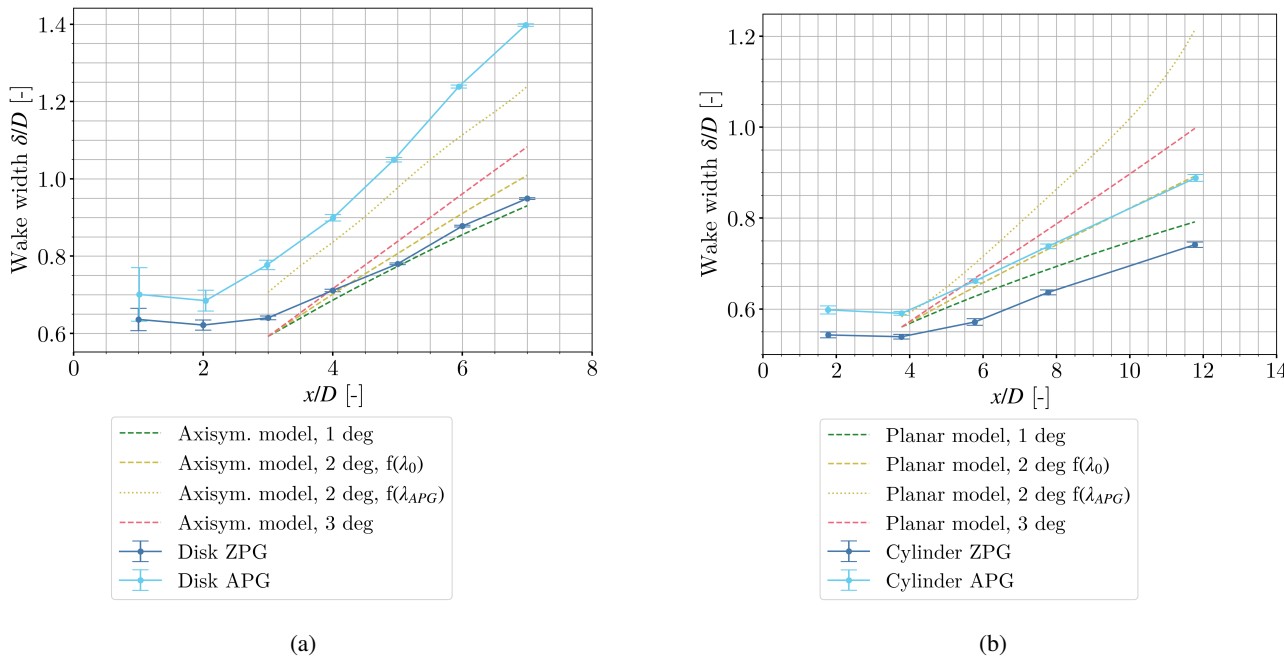

(a)  (b)

**Figure 11.** ZPG experimental results versus APG experimental results of the wake width of the wake behind the disk and cylinder. Model output with input from the ZPG case is shown for three different wall angles and a different $\lambda$. (a): Disk. (b): Cylinder.

was characterized under conditions of either no pressure gradient or an adverse one. A streamwise range of distances pertinent to wind energy applications was evaluated (2 to 12 diameters) for a porous disk and a cylinder, which are known to be

representative of horizontal and vertical axis wind turbines, respectively.

We found that the pressure gradient has a strong effect on the wake profile, centerline velocity deficit, and wake width for both families of generators. This effect is also different for the wake generated by a cylinder or a disk. Another significant different is that, the the range of Reynolds numbers studied ($2.6 \times 10^5$ to $3.9 \times 10^5$) the disk presents no Reynolds number effects, while the cylinder wake only becomes independent of this parameter for $Re_D > 2.6 \times 10^5$. Moreover, the lateral profile

of velocity deficit of the turbulent wake for both generators is properly modelled by a Gaussian curve for downstream distances larger than 4 diameters.

Within a regime that is independent of the Reynolds number based on the generator diameter, the centerline velocity deficit and wake width are significantly increased in the presence of such a gradient. This is verified for wakes that fully evolve within a pressure gradient (i.e., with the generator placed within it) or that develop through a zero pressure gradient section followed

downstream by an adverse one. Moreover, the analytical models developed by Shamsoddin and Porté-Agel, based on averaged momentum conservation, properly match all our experimental datasets.

These experiments are in good qualitative agreement with similar works on pressure gradients. The main novelty lies in the simultaneous study and comparison of planar and axisymmetric wakes within the same facility at relatively large values of turbulent Reynolds number. While most studies focus on averaged large-scale quantities, such as the velocity deficit and

370 wake width, our experimental setup can also be repurposed to study small-scale turbulence quantities, such as intermittency, dissipation, and spectral dynamics. Such studies would contribute to the development of closures for theoretical and numerical models of wind- and water-turbine generated wakes.

*Data availability.* The dataset of this manuscript is available at the CNRS "French Fluid Dynamics Database": https://entrepot.recherche. data.gouv.fr/dataverse/f2d2 under the DOI number https://doi.org/https://doi.org/10.57745/NJ15IR

*Author contributions.* WvdD acquired the data, performed the initial analysis and data investigation. WvdD and MO drafted the first draft of the manuscript. All authors developed the methodology, and reviewed and edited the manuscript.

*Competing interests.* The authors declare that they have no conflict of interest.

*Acknowledgements.* This project has received funding from the European Union's Horizon H2020 research and innovation programme under the Marie Słodowska-Curie grant agreement N°860579. We also acknowledge the assistance from Jean-Marc Breux and François Paille with

380 the mounting and running the experiments.

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
