# Peer review of "Spatial development of planar and axisymmetric wakes of porous objects under a pressure gradient: a wind tunnel study"

_Wind Energy Science, 2024_

## Referee Comment (RC2)

The manuscript is clearly written and presents interesting and generally well-documented results. I recommend the manuscript for submission provided that a few minor changes is made as described below.

L7: Please state that you only investigate positive pressure gradients - it is only for dp/dx>0 that you get deeper deficits and wider wakes (the opposite would be true for dp/dx<0)

L13-17: I think this paragraph is not entirely accurate. Strictly, you cannot predict the power output of a wind farm from the wake effect since there are several other effects that govern the power output. However, it is correct to say that it is essential to model wake effects accurately in order to predict wake losses and therefore also the total power output of a wind farm.

L78: The reference "Shamsoddin and Porte-Agel (2018)" seems to appear two times in you reference list (L382-385) so one of them should be removed.

L79: It is not correct to say inviscid since the work you refer to models a turbulent wake and turbulence inherently involves shear stresses.

L113: The parameters A, B, alpha and beta are related via momentum conservation, but you fit them as independent parameters. Why not make the fit while ensuring that they are still related in the right way? Does your approach imply that your wake profile does not fulfill momentum conservation? It would be good to include a sentence about this in the manuscript.

 L180: In the work by Neunaber et al. (2021) the reported drag coefficient includes the drag of the tower. In your work, you have no tower so one should expect a lower drag coefficient than what is reported by Neunaber et al. (2021). Have you measured the drag coefficient to confirm that it is indeed what you expect?

L215: increasing is misspelled

L230-: You could consider writing that some of the differences you observe between disc and cylinder is also reflecting that the cylinder is essentially a 2D flow case while the disk is more 3D. Generally, 2D bodies produce deeper wake than 3D bodies.

L240: Why is wake of the cylinder skewed? Is it lack of statistical convergence or is the flow in the tunnel asymmetric?

Caption figure 5 : "Radial velocity profiles" sounds like it is the radial velocity and not the streamwise velocity. What you mean is something like "Radial (or horizontal) profiles of the (streamwise) velocity.

L270-272: You mention that the best fit is obtained at an angle of 3 degrees and that this is not consistent with the best fit in the empty tunnel. You mention several reasons for this, but it could maybe also be due to uncertainties in the thrust/drag coefficient (which is not measured) or what?

L289-L290: You write that "We remark that for this case, given the limitations of the experimental setup, the case where the wake evolves both across the test and the diffuser sections was not considered". However, Figure 1 indicates something different – namely that you did perform tests with the cylinder in the test section. Am I misunderstanding something?

L305-308: To state that there is no Reynolds number effect is not entirely accurate when looking at Figure 7. I would say that there is a low sensitivity to Reynolds number.

---

## Author Comment (AC1)

**Author Response to Reviews of**

**Spatial development of planar and axisymmetric wakes of porous objects under a pressure gradient: a wind tunnel study**

Wessel van der Deijl, Martin Obligado, Stéphane Barre and Christophe Sicot

**RC:** Referee Comment,     AC: *Author Comment*

**Referee 1**

**1. General considerations**

**RC:** The article presents an experimental study of porous disk and cylinder wakes under zero and adverse pressure gradients. Moreover, the experimental data is compared with predictions of analytical wake models under pressure gradient for planar and axisymmetric wakes. The findings of the study are not necessarily new; however, they are consistent with and support the findings of several previous experimental and numerical studies on the topic. As argued by the authors, it is the first study that investigates both planar and axisymmetric wakes under comparable flow conditions in the same test facility. The study can be useful for the community and I will recommend its publication once the concerns highlighted in the following sections are addressed.

**AC:** *We thank the reviewer for their positive comments and the time invested in the present review. We are glad to see that they appreciated our manuscript. We are also glad that they recommend its publication, once we address the comments highlighted below. Major changes done to the main manuscript are highlighted in blue.*

**2. Specific comments:**

**RC:** 1. The authors claim that the cylinder wake matches the wake of a vertical axis turbine. However, the cylinder used in the study is a two-dimensional cylinder resulting in a planar wake (as suggested by the title of the article). The wake of a vertical axis wind turbine, on

the other hand, is not a planar wake. Maybe for very large aspect ratios, it could be similar to a planar wake, but it cannot be true in general. The authors need to clarify this and specify under which conditions the wake of a vertical axis turbine matches with a cylinder (planar) wake.

AC: *The referee raises a fair point, as we only studied the planar wake, which is not, in general, the same as the wake of a VAWT. Moreover, whether the wake of a cylinder matches that of the wake of a VAWT, strongly depends on the solidity of the VAWT. The higher the solidity, the higher the resemblance of a bluff body wake (Araya, Colonius, and Dabiri, 2017). Furthermore, even if the aspect ratio and the solidity match, the VAWT wake is only similar to that of a cylinder in the far wake. Finally, the aspect ratio is important in the wake recovery process for a VAWT. The vertical convection (which would be out-of-plane for this planar wake) and $x - z$ Reynolds stresses play a dominant role (Araya, Colonius, and Dabiri, 2017, Boudreau and Dumas, 2017), likely more than the in-plane recovery processes (Deijl et al., 2022). Therefore, ultimately the planar wake studied in this work and its model would be a poor match for some streamwise ranges of a VAWT. Nevertheless, because of this dependency on aspect ratio, the edge case of an infinite aspect ratio planar wake is still relevant. Consequently, it is still important to model the planar wake generated by a porous cylinder.*

AC: *In the new manuscript, a sentence has been added in the introduction to include the fact that the cylinder wake is not the same as an actual VAWT wake.*

RC: 2. How is the theoretical baseline velocity estimated? The authors must provide the equation used for it.

AC: *We thank the reviewer for remarking this omission in our manuscript. The tunnel area at $x$ is given by:*

$$A(x) = (H + 2x \tan{(\gamma)})(W + 2x \tan{(\gamma)}), \qquad (1)$$

AC: *with $x$ being the distance from the start of the divergent section and $\gamma$ the angle of the walls. The height of the test section of the tunnel $H$ is 2.6 m and the width of the test section of the tunnel $W$ equals 2.4 m. The expected baseline velocity $U_b$ at $x$ is then related to the measured velocity in the test section $U_\infty$ by:*

$$\frac{U_b(x)}{U_\infty} = \frac{A(0)}{A(x)}. \qquad (2)$$

AC: *This equation has now been added to section 3.*

RC: 3. Page 10 lines 221-223, the authors state that they have verified that the ratio of velocity deficit and wake width is same under zero and adverse pressure gradients, however, they have not included a supporting figure. Is it excluded for brevity? If so, please specify.

AC: *It was indeed excluded for the sake of brevity. It is now added in the manuscript and shown in figure 1 in this rebuttal as it is indeed a key point in the model. From this figure it follows that, for a given wake generator, the ratio of $U_b(x)C(x)/\delta(x)$ has a very similar slope, regardless of the pressure gradient. Figure 1 corresponds therefore well to the results of Liu (Liu, Thomas, and Nelson, 2002) and Thomas and Liu (Thomas and Liu, 2004), whose work form the basis of this assumption. However, small variations (<20%) between the APG and the ZPG cases remain, which is ultimately important for calculating the expected wake width. In the work of Liu (Liu, Thomas, and Nelson, 2002) and Thomas and Liu (Thomas and Liu, 2004), these absolute differences were also present and they were attributed to changes in Re-number. Due to the different set-ups for the ZPG and APG cases (such as a slightly different blockage ratio), small absolute differences between the wakes are therefore unavoidable. For the calculation of the differential equations of equation 2 and equation 9 of the main manuscript, these absolute variations are negligible and most parameters can be non-dimensionalised. Nevertheless, an absolute difference of $\lambda$ of around 20% may result in a significant change in the calculated wake width, as it can be deduced from equation 4 of the main manuscript. To further show this, the error due to $\lambda$ has been added to the model results.*

AC: *Figure 1 frim this rebuttal has been added to the main manuscript as figure 7. A discussion on $\lambda$ has been added to the manuscript (section 4.2). A recommendation for future work has been added to the conclusions regarding the estimation of $\lambda$.*

RC: 4. The authors show self-similarity of wake deficit in figure 5c and 5d. According to the data, for porous disks, the wake is self-similar even at x=1D. Why is that? I believe this could be related to the geometry of the disk. In principle, for a turbine wake the self-similarity is valid in the far wake, which scales with turbulence intensity (the lower the turbulence intensity, the further from the turbine is the onset of the far wake). Given the very low turbulence intensity (0.25%), it is surprising to see self-similarity at x=1D.

[Figure]

Figure 1: $\lambda(x) = U_b(x)C(x)/\delta(x)$, for all generators at $Re_D = 3.9 \times 10^5$.

Perhaps, the authors can explain this in more detail.

AC: *We thank again the reviewer, as this comment made us find a problem with the presentation of our results. Indeed, in the legend of figures 5c and d from the previous version of the manuscript, the entry for $x = 1D$ is included, but those lines are not present in figure 5c. As can be seen in figure 5a, the velocities in the wake at $x = 1D$ (and also $x = 2D$ for the APG case) appear to become negative and could not be registered by the pressure sensors. Therefore, the self-similarity could not be tested because of the unknown maximum velocity deficit. These profiles were not included in figure 5c, but erroneously appeared in the legend. The figure has been corrected, to figure 2 below. Still, for the ZPG case, the profile at $x = 2D$ is already quite similar to the other profiles. This is indeed likely due to the geometry of the porous disk, which is axisymmetric and the porosity is evenly spread, in contrast with a turbine with a fixed number of blades. In the new manuscript, figure 5 and the corresponding discussion have been modified accordingly to this discussion.*

RC: 5. In section 5, the authors use power law fit (equations 5 and 6) to model the ZPG wake. These equations have 5 parameters ($A$, $\alpha$, $B$, $\beta$, ). It appears to me that for each case (either porous disk or cylinder), the authors have directly fitted the equations to the experimental data resulting in 5 fitted parameters. My first question is how general or universal are these parameters and are they valid only for the cases presented in this study? Secondly, if you

[Figure]

Figure 2: Velocity profiles normalized using the centreline velocity deficit $\Delta u(x)$ and the wake width $\delta$ for the disk.

need the experimental wake under ZPG to fit these parameters (i.e. the ZPG wake needs to be known a-priori), then why not directly used the experimental wake under ZPG for the input of the pressure gradient models? Why go through an extra step to fit power law equations to the ZPG wake and then use it for the pressure gradient model?

AC: *The referee is correct that this step is, in principle, not necessary. One should only need the parameters of the ZPG wake and the pressure gradient as input to the model. However, because of the relatively low spatial resolution of the experimental wake, a smooth fit has to be made to feed into the differential equation solver. While this could have been achieved by a simple cubic spline or any fitting function, the power laws that were chosen as fitting function are based on the theory by Townsend (Townsend, 1976) and George (George, 1989) from which the streamwise evolution of the velocity deficit and wake width in the wake of a bluff-body are derived. The Townsend-George theory also provides the actual values of the $\alpha$ and $\beta$ parameters (the other three values are scaling factors and a change of the power law origin). This theory assumes an axisymmetric self-similar wake and the classic Richardson-Kolmogorov energy cascade. Whether these power laws apply in general to wind turbine wakes is a topic of itself (Neunaber, Peinke, and Obligado, 2022), but they have previously been applied to wind turbine wake models (Porté-Agel, Bastankhah, and*

*Shamsoddin, 2020). The universality of the fitting parameters is still under debate, as they have been found to depend not only on the generator bu also on the background turbulence level and the tip-speed ratio of the rotor (see the recent work of Bourhis and Buxton, 2024 for a detailed discussion on this point).*

**AC:** *Therefore, by assuming such power laws for the streamwise evolution of averaged quantities, a smooth function was created as input to the ODE, that still satisfies the expected theoretical streamwise evolution. By tuning the values of the power laws to the data, we ensure a better fit than the theoretical values such that the focus is on the pressure gradient model performance, instead of the fit of these power laws. Nevertheless, the exact values that have been used in this case are not applicable in a general sense.*

**AC:** *In conclusion: for a general application of the model, it depends on the quality and resolution of the input whether this step is required. Ideally, if the data is smooth enough, it could directly be fed into an ODE solver. If it is not, we think that a power law is a suitable fit for the streamwise evolution of the velocity deficit and the wake width.*

**AC:** *A clarification on the application of these power-law-based scalings to the data has been added to section 2.1.*

**RC:** 6. Page 16 lines 276-277, how do the authors estimate that the far wake starts from $x/D \leq 3$? In addition, how practical is it to pick the velocity deficit at the start of the far wake from the experimental data?

**AC:** *The reason for the estimation of the onset of the far wake at $x/D \geq 3$ is two-fold. First of all, from a practical stand-point, the measured velocities in the wake of the disk showed negative velocities at $x = 1D$ and $x = 2D$ for the APG case. The exact velocity deficit and wake width were therefore difficult to estimate and to use in the model. For $x/D \geq 3$, this problem vanishes. Second of all, from the self-similarity figures 5c-d, it appears that the profiles for $x/D \geq 3$ are all self-similar, which is a requirement for the model. At $x/D = 2$, for both the disk but especially the cylinder, the profile deviates slightly from the other profiles. In consequence, we consider that $x/D = 2$ could still be considered within the near-wake range, at least from an averaged velocity perspective.*

**AC:** *Regarding the second point about the practicality of picking the velocity deficit from the experimental data, we agree that this may not be indeed ideal, as it means that one would need information a priori at a certain point about the APG wake before modelling it from*

*that starting point. Dar and Porté-Agel (Dar and Porté-Agel, 2022) propose the equation shown in section 2.2 to estimate the velocity deficit in the near wake, which can be used instead. However, this does add an extra assumption to the model, that sums up to the unknown velocity deficit in the near wake in the case of the disk. In consequence, we chose to use the velocity deficit that was known. This was done to focus on the predicted evolution of the deficit by the model.*

**AC:** *A brief sentence in section 2.2. has been added to clarify that this starting point is where the wake is deemed self-similar, which is approximately for $x/D \geq 3$.*

**RC:** 7. It would be useful if the authors add a comparison of the full velocity deficit profiles between the experiments and the model.

**AC:** *This is indeed a good suggestion. This comment actually highlighted the model's sensitivity to $\lambda$ of equation 4 mentioned above. As discussed in comment 3, the ratio of velocity deficit and wake width is not exact in an absolute sense between the ZPG wake and the APG wake. While the model has a robust prediction of the velocity deficit, the wake width is relatively sensitive to this ratio $\lambda$. This is now highlighted in the new manuscript by adding the model output of the wake width as figure 11. As can be seen from these figures, the wake width has a larger error due to $\lambda$. We therefore recommend that in future work the relation between the velocity deficit and the wake width is calculated in a more robust sense, for instance by using the full profiles and not only the centreline evolution.*

**AC:** *We have added figure 11 showing the wake width output of the model, visualizing the sensitivity of the model output of velocity deficit and wake width to $\lambda$. We have also added a discussion on $\lambda$ in section 4 and added a recommendation to the conclusions. We have not included the full velocity profiles, as figures 9, 10 and 11 show the velocity deficit and the wake width output and combined they constitute the same velocity profiles.*

**3. Technical comments**

**RC:** Page 3 line 65, the averaged streamwise velocity 'deficit' is self-similar in the far-wake, please correct.

**AC:** *'Deficit' has been added to this sentence.*

**RC:** Why is the velocity deficit plotted as $(1 - C(x))$? Is there a specific reason for this?

**AC:** *Showing $1 - C(x)$ or $C(x)$ is ultimately the same. We think that it might be slightly easier to see the asymptote to 1 than the asymptote to 0 in these graphs.*

**AC:** *We thank again the reviewer for this review and the valuable comments and suggestions they made. We hope they will find the new version of the manuscript suitable for publication in Wind Energy Science.*

**References**

Araya, D.B., T. Colonius, and J.O. Dabiri (2017), Transition to bluff-body dynamics in the wake of vertical-axis wind turbines, in: *Journal of Fluid Mechanics* 813, pp. 346–381, ISSN: 14697645.

Boudreau, Matthieu and Guy Dumas (2017), Comparison of the wake recovery of the axial- flow and cross- flow turbine concepts, in: *Journal of Wind Engineering and Industrial Aerodynamics* 165.March, pp. 137–152.

Deijl, W. van der et al. (2022), Experimental study of mean and turbulent velocity fields in the wake of a twin-rotor vertical axis wind turbine, in: *Journal of Physics: Conference Series* 2265.2, p. 022073, ISSN: 1742-6596.

Liu, Xiaofeng, Flint O. Thomas, and Robert C. Nelson (2002), An experimental investigation of the planar turbulent wake in constant pressure gradient, in: *Physics of Fluids* 14.8, pp. 2817–2838, ISSN: 10706631.

Thomas, Flint O. and Xiaofeng Liu (2004), An experimental investigation of symmetric and asymmetric turbulent wake development in pressure gradient, in: *Physics of Fluids* 16.5, pp. 1725–1745, ISSN: 10706631.

Townsend, A A (1976), *The structure of turbulent shear flow*, Cambridge University Press.

George, W K (1989), The self-preservation of turbulent flows and its relation to initial conditions and coherent structures, in: *Advances in Turbulence*, ed. by W K George and R Arndt, Springer.

Neunaber, Ingrid, Joachim Peinke, and Martin Obligado (2022), Application of the Townsend–George theory for free shear flows to single and double wind turbine wakes – a wind tunnel study, in: *Wind energy science*, 7, Copernicus GmbH, pp. 201–219.

Porté-Agel, Fernando, Majid Bastankhah, and Sina Shamsoddin (2020), Wind-Turbine and Wind-Farm Flows: A Review, in: *Boundary-Layer Meteorology* 174, pp. 1–59.

Bourhis, M and ORH Buxton (2024), Influence of freestream turbulence and porosity on porous disk-generated wakes, in: *Physical Review Fluids* 9.12, p. 124501.

Dar, A.S. and F. Porté-Agel (2022), An Analytical Model for Wind Turbine Wakes under Pressure Gradient, in: *Energies 2022, Vol. 15, Page 5345* 15.15, p. 5345, ISSN: 1996-1073.

---

## Author Comment (AC2)

**Author Response to Reviews of**

**Spatial development of planar and axisymmetric wakes of porous objects under a pressure gradient: a wind tunnel study**

Wessel van der Deijl, Martin Obligado, Stéphane Barre and Christophe Sicot

**RC:** Referee Comment,    AC: *Author Comment*

**Referee 1**

**1. General considerations**

**RC:** The manuscript is clearly written and presents interesting and generally well-documented results. I recommend the manuscript for submission provided that a few minor changes is made as described below.

*AC:* *We thank the referee for the time they have invested in this review and we are very glad to hear that they found the results interesting. We will address the referee's comments below. Major changes done to the main manuscript are highlighted in blue.*

**2. Specific comments:**

**RC:** L7: Please state that you only investigate positive pressure gradients - it is only for dp/dx>0 that you get deeper deficits and wider wakes (the opposite would be true for dp/dx<0)

*AC:* *'Adverse' has now been added to the sentence in line 7. It is indeed true that only an adverse pressure gradient results in deeper deficits and wider wakes.*

**RC:** L13-17: I think this paragraph is not entirely accurate. Strictly, you cannot predict the power output of a wind farm from the wake effect since there are several other effects that govern the power output. However, it is correct to say that it is essential to model wake effects accurately in order to predict wake losses and therefore also the total power output of a wind farm.

**AC:** *We thank the reviewer for this remark. In the new manuscript, this paragraph has been rewritten to focus on the relevance of turbulence and wind turbine farms. On the one side, we now discuss the overall interaction between wind farms and the atmospheric boundary layer, which governs the entrainment of energy. On the other side, we also remark the relevance of the wake effects and wake losses.*

**RC:** L78: The reference "Shamsoddin and Porte-Agel (2018)" seems to appear two times in your reference list (L382-385) so one of them should be removed.

**AC:** *Corrected.*

**RC:** L79: It is not correct to say inviscid since the work you refer to models a turbulent wake and turbulence inherently involves shear stresses.

**AC:** *The sentence has been slightly rewritten. The flow is not inviscid, but the viscous terms (among others) have been neglected from the conservation of momentum equation for a turbulent flow to derive the ODE.*

**RC:** L113: The parameters A, B, alpha and beta are related via momentum conservation, but you fit them as independent parameters. Why not make the fit while ensuring that they are still related in the right way? Does your approach imply that your wake profile does not fulfill momentum conservation? It would be good to include a sentence about this in the manuscript.

**AC:** *The reviewer is correct, and a 3-parameter fit (using the virtual origin and one exponent and one pre-factor) could be applied to adjust simultaneously the velocity deficit and wake width. Nevertheless, most of experimental works on turbulent wakes follow our approach and perform a 5-parameter fit (see Nedic et al. PRL (2013) and references therein). The reason is that the resolution in terms of streamwise positions need to be too high to guarantee high-quality fitting, and it is therefore better to fit both quantities independently while checking, a posteriori, their consistency in terms of momentum conservation. In this way, momentum conservation is assumed to be a property of the wake and verified using the output of the 5-parameter fit A small discussion on this point has been added to section 2.*

**RC:** L180: In the work by Neunaber et al. (2021) the reported drag coefficient includes the drag of the tower. In your work, you have no tower so one should expect a lower drag coefficient than what is reported by Neunaber et al. (2021). Have you measured the drag coefficient to

confirm that it is indeed what you expect?

**AC:** *The reviewer raises a fair point. Indeed, the drag coefficient in our case should be slightly lower than the one of the cited work. Unfortunately, our experimental setup does not allow to measure the drag coefficient of the disk so we cannot confirm this value. In the work by Neunaber et al. (2021), the tower would increase the frontal area within the radius of the disk by at most 3.8% (since the tower has a 3.8cm diameter and the disk 59cm). If the total drag coefficient is proportional to this frontal area, that would mean that the drag coefficient, without the tower, would be around 0.92. Following this remark, in the new version of the manuscript, the thrust coefficient has been changed to $C_T \approx 0.9$. The same has been done for the thrust coefficient of the cylinder. $C_T = 0.9$ better reflects the accuracy of the approximation of the thrust coefficient.*

**RC:** L215: increasing is misspelled.

**AC:** *Corrected.*

**RC:** L230-: You could consider writing that some of the differences you observe between disc and cylinder is also reflecting that the cylinder is essentially a 2D flow case while the disk is more 3D. Generally, 2D bodies produce deeper wake than 3D bodies.

**AC:** *This is a good suggestion. A few sentences have been added about the general differences between the two flow cases.*

**RC:** L240: Why is wake of the cylinder skewed? Is it lack of statistical convergence or is the flow in the tunnel asymmetric?

**AC:** *The baseline velocity in the tunnel has been measured and for an empty test section the flow was found to be relatively symmetric. At least, no strong asymmetries to the extent that they would show up in the wake have been found, especially in the test section (the ZPG case). What could be possible is that the cylinder was positioned at a very small angle, such that the porous holes were not completely symmetrically aligned with the incoming flow. A sentence about this skewed wake has been added to the new version of the manuscript (section 4.1).*

**RC:** Caption figure 5 : "Radial velocity profiles" sounds like it is the radial velocity and not the streamwise velocity. What you mean is something like "Radial (or horizontal) profiles of the (streamwise) velocity.

AC: *The caption has been changed to the suggested horizontal profiles'.*

RC: L270-272: You mention that the best fit is obtained at an angle of 3 degrees and that this is not consistent with the best fit in the empty tunnel. You mention several reasons for this, but it could maybe also be due to uncertainties in the thrust/drag coefficient (which is not measured) or what?

AC: *The exact value of the thrust coefficient would be important if one would try to calculate the strength of the wake from the thrust coefficient alone. The model has however been applied to a measured value of the velocity deficit in the wake, after which the evolution of the wake is calculated (in a pressure gradient). If the thrust coefficient changes, this starting velocity deficit would have changed, so the starting point of the model would have also changed. After this point, the thrust coefficient is no longer an input in the model. Furthermore, the thrust coefficient would have changed the strength of the wake in both the ZPG and the APG cases, and the evolution of the ZPG wake is an input to the APG case. In consequence, we think that an uncertainty in the thrust coefficient is not a factor in the outcome of the model.*

RC: L289-L290: You write that "We remark that for this case, given the limitations of the experimental setup, the case where the wake evolves both across the test and the diffuser sections was not considered". However, Figure 1 indicates something different – namely that you did perform tests with the cylinder in the test section. Am I misunderstanding something?

AC: *Three cases have been considered and measured. The ZPG case, the APG case and the ZPG case that continues into an APG. The ZPG case and the APG case were done for both the disk and the cylinder, while for the disk we also covered the case where the disk was placed in the ZPG and measurements extended into an APG case. This was not done for the cylinder. The sentence has been slightly rewritten in the new manuscript to clarify this.*

RC: L305-308: To state that there is no Reynolds number effect is not entirely accurate when looking at Figure 7. I would say that there is a low sensitivity to Reynolds number.

AC: *We agree with this remark and no Reynolds number effects' has been changed to low sensitivity to Reynolds number'.*

AC: *We would like to sincerely thank the referee again for their time and for providing this*

*valuable review. We believe that the referee's comments have improved the article and we hope that the article can be considered for publication in Wind Energy Science.*

---

## Author Response (AR2)

**Replies Reviewer #1 for the manuscript titled:**
**Spatial development of planar and axisymmetric wakes of porous objects under a pressure gradient: a wind tunnel study**

Wessel van der Deijl, Martín Obligado, Stéphane Barre & Christophe Sicot

February 6, 2025

*I'm generally satisfied with the answer and corrections made by the authors and therefore recommend the manuscript for publication. I only have one minor suggestion for change: namely to show figure 7 as lambda/lambda0 and thereby document that the APG and ZPG case collapse as state line line 260-261. In addition to this figure, the authors could provide a table or another figure which shows lambda0 to give the full picture.*

We thank again the reviewer for investing time in this review. Following their advice, we have now added a new panel in figure 7, and rewritten and expanded section 4.2 that discusses the trends observed for the constant $\lambda$. In the new manuscript, significant changes are highlighted in blue.